Potency of bisresorcinol from Heliciopsis terminalis on skin aging: in vitro bioactivities and molecular interactions

Saechan Charinrat 1
Nguyen Uyen Hoang 2
Wang Zhichao 2
Sugimoto Sachiko 2
Yamano Yoshi 2
Matsunami Katsuyoshi 2
Otsuka Hideaki 3
http://orcid.org/0000-0003-4989-4183 Phan Giang Minh 4
http://orcid.org/0000-0002-9016-0694 Pham Viet Hung 5
Tipmanee Varomyalin 6 tvaromya@medicine.psu.ac.th
Kaewsrichan Jasadee 1 jasadee.k@psu.ac.th
1 Department of Pharmaceutical Chemistry and Drug Delivery System Excellence Center, Faculty of Pharmaceutical Sciences, Prince of Songkla University , Songkhla , Thailand
2 Graduate School of Biomedical and Health Sciences, Hiroshima University , Hiroshima , Japan
3 Faculty of Pharmacy, Yasuda Women’s University , Hiroshima , Japan
4 Faculty of Chemistry, VNU University of Science, Vietnam National University , Hanoi , Vietnam
5 Research Center for Environmental Technology and Sustainable Development, VNU University of Science, Vietnam National University , Hanoi , Vietnam
6 Department of Biomedical Sciences and Biomedical Engineering, Faculty of Medicine, Prince of Songkla University , Songkhla , Thailand
Park Eun-Jung
Electronic publication date: 2021 Jun 22
Publication date: 2021
Volume: 9
Electronic Location ID: e11618
Received 2021 Feb 26; Accepted 2021 May 25
Copyright: © 2021 Saechan et al.
Copyright year: 2021
Copyright holder: Saechan et al.
License: This is an open access article distributed under the terms of the Creative Commons Attribution License, which permits unrestricted use, distribution, reproduction and adaptation in any medium and for any purpose provided that it is properly attributed. For attribution, the original author(s), title, publication source (PeerJ) and either DOI or URL of the article must be cited.
License URL: https://creativecommons.org/licenses/by/4.0/

Keywords: Heliciopsis terminalis, Bisresorcinol, Collagenase, Elastase, Tyrosinase, Anti-aging property, Molecular docking, Enzyme inhibition

Funding: The Royal Golden Jubilee PHD/0151/2556 Thailand Research Fund (TRF) Thailand, and the National Research Council of Thailand (NRCT) This research project was supported financially by The Royal Golden Jubilee Ph.D. Program (PHD/0151/2556), the Thailand Research Fund (TRF), Thailand, and the National Research Council of Thailand (NRCT). The funders had no role in study design, data collection and analysis, decision to publish, or preparation of the manuscript.

==============================
Background

A bisresorcinol was isolated as the main constituent of Heliciopsis terminalis’s trunk (Proteaceae). Recently, resorcinol is applied as an active whitening agent in various cosmetic products. Because of the structural mimic to resorcinol, benefits of the bisresorcinol as an aging-enzyme antagonist were demonstrated in this study.

Methods

The bisresorcinol was purified from the crude ethanolic extract of H. terminalis’s trunk by solvent extraction and preparative chromatography, respectively. Inhibitory activity on collagenase, elastase, and tyrosinase of the compound was investigated by using a different spectroscopic technique. Molecular docking was carried out to predict possible interactions of the substance around the enzyme active sites.

Results

The IC50 values on collagenase of the bisresorcinol and caffeic acid were 156.7 ± 0.7 and 308.9 ± 1.6 µmole L−1, respectively. For elastase activity, the IC50 of 33.2 ± 0.5 and 34.3 ± 0.3 µmole L−1 was respectively determined for the bisresorcinol and ursolic acid. The bisresorcinol was inhibitory to tyrosinase by exhibiting the IC50 of 22.8 µmole L−1, and that of 78.4 µmole L−1 was present for β-arbutin. The bisresorcinol bound to collagenase, elastase, and tyrosinase with the respective binding energies of −5.89, −5.69, and −6.57 kcal mol−1. These binding energies were in the same ranges of tested inhibitors. The aromatic phenol groups in the structure were responsible for principle as well as supporting binding interactions with enzymes. Hydrogen binding due to hydroxyl groups and π-related attractive forces from an aromatic ring(s) provided binding versatility to bisresorcinol.

Conclusion

The bisresorcinol purified from H. terminalis might be useful for inclusion in cosmetic products as an aging-enzyme antagonist.

Introduction

A bisresorcinol, (8′Z)-3,5-dihydroxy-1-[16′-(3″,5″-dihydroxyphenyl)-8′-hexadecen-1′-yl]benzene, has been isolated as an abundant constituent from the trunk of Heliciopsis terminalis (Kurz) Sleumer, which is a plant of Proteaceae family and has been used by ethnic people in Vietnam for hepatoprotective effect. To date, medical benefits of this compound as an agent for anti-inflammation, inhibition of malondialdehyde (MDA) production, and hepatoprotection, have been clarified in vitro and in vivo (Giang et al., 2019). It is apparent in that a molecule of the bisresorcinol contains two resorcinol molecules (EC No 203-585-2), bonded together with meta-C7H14C = CC7H14 linkage (Fig. 1A). Resorcinol is recently applied in a form of facial concentrated serum for white therapy. Based on the Scientific Committee on Consumer Safety, Brussels, Belgium (SCCS 1270/09, European Commission, Scientific Committee on Consumer Safety, 2010), resorcinol has been regulated as an oxidant in products intended for coloring eyelashes at concentration ranges up to 1.25%, or in hair lotions and shampoos at concentration ranges up to 0.5%. Cosmetical applications of resorcinol are thus diverse. For whitening agents, inhibition of melanin synthesis by melanocytes that reside in the basal cell layer of epidermis has been assumed. Tyrosinase (EC 1.14.18.1) is a key enzyme for catalyzing melanogenesis in which L-tyrosine is used as the substrate. Although melanin helps protect skin form damage of UV rays, excess melanin production can cause hyperpigmentation, freckle, and age spot. In addition to resorcinol, various synthetic and natural compounds with distinct chemical structures have been demonstrated to exhibit anti-tyrosinase activity (Dobos et al., 2015).

Figure 1 Chemical structures of compounds.

(A) Bisresorcinol isolated from H. terminalis’s trunk; (B) caffeic acid; (C) ursolic acid; and (D) β-arbutin.

UV radiation is a major external factor that causes skin aging. Instead, skins can age over time by reactive oxygen species (ROS) and lipid peroxides that are internally produced after UV exposure. Secondary products from lipid peroxide metabolisms can damage elastin and collagen fibers of extracellular matrices (ECMs). Therefore, exposure to UV rays can result in dehydration, loss of elasticity, as well as wrinkles of the skins. To our knowledge, the ECMs are periodically remodeled to maintain normal tissue structures in accompany with that expired tissue proteins are properly degraded by matrix metalloproteinases, such as collagenases, and serine proteases like elastases (Thring, Hili & Naughton, 2009). In addition, both quantities and activities of aging enzymes have been increased by oxidative stress, leading to extensive ECMs degradation and progressive aging (Sherratt et al., 2019). Therefore, compounds that can form complexes with metal ions are anticipated to be inhibitory to collagenases, elastases, and tyrosinase accordingly (Selvaraj et al., 2014). This project aims to determine biological effects of the bisresorcinol on tyrosinase, collagenase, and elastase in vitro, and docking technique was carried out to clarify their interactions at molecular levels. It is hopeful to delay the aging process and improve skin appearance by the bisresorcinol.

Materials & methods

Chemicals and instruments

All the chemicals used in this study were of analytical grade. Organic solvents, including 1-butanol, acetone, chloroform (also for NMR), and ethyl acetate, were purchased from KANTO CHEMICAL CO., INC., Japan. Chemicals, such as hydrochloric acid, K2HPO4, KH2PO4, L-tyrosine, silica gel, octadecyl silica gel, and tris (hydroxymethyl) aminomethane, were bought from NACALAI TESQUE, INC., Japan. Standard substances, including β-arbutin, caffeic acid, and ursolic acid, and tricine were obtained from Tokyo Chemical Industry, Japan. Enzymes, such as collagenase, elastase, and tyrosinase, including N-succinyl-Ala-Ala-Ala-p-nitroanilide (SANA) and dimethyl sulfoxide (DMSO), were acquired from Sigma-Aldrich, Germany. DMSO and methanol (NMR grade) were purchased from ISOTEC®, Sigma-Aldrich, Germany. A peptide: MOCAc-PRO-Leu-Gly-Leu-A2pr(Dnp)-Ala-Arg-NH2, was ordered from Peptide Institute, Inc., Japan. Instruments used included HPLC (SCL-10A sp/RID-6A/c-R3A; SHIMADZU, Kyoto, Japan), Fluorescence microplate reader (Enspire 2300 Multimode reader; PerkinElmer, Waltham, MA, USA), HR-ESI-MS (LTQ orbitrap XL; Thermo Fisher Scientific, Waltham, MA, USA), Infrared Spectrophotometer (FT-720, HORIBA), Nuclear Magnetic Resonance Spectrometer (ECA-500/600, JEOL), and UV-Vis microplate reader (Multiskan Go; Thermo Scientific, Waltham, MA, USA).

Compound purification, identification and ratification

The crude extract in ethanol of H. terminalis’s trunk was generously obtained from Professor P.M. Giang, Faculty of Chemistry, Vietnam National University, Hanoi, Vietnam with voucher specimen number of HNIP-18473. Then, it was partitioned in methanol/n-hexane by three times repeated. The methanol layer was evaporated to nearly dry and subsequently partitioned in ethyl acetate/1-butane. The ethyl acetate layer was vaporized until its volume was reduced by 80%. The resulting viscous liquid was then purified on silica gel column using chloroform/methanol gradient elution as follows: 100% chloroform, 30:1, 20:1, 10:1, 7:1, 5:1, 3:1, 3:2, and 2:1 of chloroform/methanol, and 100% methanol, respectively. The eluent from the 5:1 mixed solvent was collected and evaporated to one fourths volume, followed by the separation on the ODS column using gradient elution of methanol and acetone. The desired compound, i.e., bisresorcinol (Fig. 1), was obtained from eluents containing 70% acetone and confirmed by MS, NMR and IR spectroscopies in regard to the instrumental library data. Thin layer chromatography was used for estimation of its purity.

Biological activity assays

In vitro experiments that consisted of enzyme-inhibitory assays on collagenase, elastase and tyrosinase for the bisresorcinol and known inhibitors were carried out as follows.

Collagenase inhibition

The recent anti-collagenase assay was modified from that previously described by Widyowati et al. (2016). In brief, 50 mmol L−1 tricine buffer pH 7.5 supplemented with 400 mmol L−1 NaCl and 10 mmol L−1 CaCl2 was used as the buffering diluent. Collagenase from Clostridium histolyticum (EC.3.4.24.3) was dissolved in the buffer to a concentration of 1 mg L−1. The enzyme substrate, MOCAc-PRO-Leu-Gly-Leu-A2pr (Dnp)-Ala-Arg-NH2, was prior dissolved in DMSO and subsequently diluted in the buffer to a concentration of 1 mmol L−1. A sample dissolved in DMSO was diluted to different concentrations by using the buffer. A 2-µl sample solution was incubated with 100 µl enzyme solution at 37 °C for 10 min. Then, a 50-µl substrate was added and thoroughly mixed. Fluorescence emission (F) at 405 nm was immediately recorded and continually monitored for 30 min using a wavelength of 320 nm for excitation. Caffeic acid was used as a position control (Fig. 1). Negative control was performed with water. The percent inhibition (%) was calculated from the equation below:

%Inhibition={1–(Fsam,30–Fsam,0)÷(Fcont,30–Fcont,0)}×100.

Elastase inhibition

The assay according to Abhijit & Manjushree (2010) was applied with some modifications. Briefly, the buffer system was 0.2 mM Tris-HCl buffer (pH 8.0). A solution of 1 µg mL−1 porcine pancreatic elastase (EC.3.4.21.36) was prepared in sterile water. The enzyme substrate, N-Succinyl-Ala-Ala-Ala-p-nitroanilide (SANA), was dissolved in the buffer to a concentration of 80 mmol L−1. A sample dissolved in DMSO was diluted to various concentrations by using the buffer. A 100-µl sample solution was mixed with 50 µl enzyme solution and incubated for 15 min. After that 50 µl of the enzyme substrate solution was added and thoroughly mixed. The OD410 was measured immediately (0 min) and after incubation overnight (o/n) at 37 °C by using a microplate reader. Ursolic acid was used as a positive control (Fig. 1), and water was used as a negative control. The percentage inhibition (%) was calculated by the following equation:

%Inhibition={1–[(Asam,o/n–Asam,0)÷(Acont,o/n–Acont,0)]}×100.

Tyrosinase inhibition

The used tyrosinase inhibitory assay was adopted from the previously described method (Jiratchayamaethasakul et al., 2020). In brief, the assay was carried out in 0.05 mol L−1 phosphate buffer pH 6.8. Mushroom tyrosinase (EC.1.14.18.1) was dissolved in the buffer to a concentration of 100 units mL−1. L-tyrosine, an enzyme substrate, was prepared to a concentration of 0.25 mg mL−1 in the buffer. A test compound dissolved in DMSO was diluted to concentrations ranging between 0.625 and 10 mg mL−1. Into a well of 96-well plates, 10 µl sample and 40 µl L-tyrosine solution were mixed and incubated for 10 min at room temperature. After that 50 µl enzyme solution was inoculated and mixed thoroughly. The reaction was stopped afterward by incubation on ice for 1 min. The OD475 was measured using a microplate reader. β-Arbutin and water were used as a positive control (Fig. 1) and a negative counterpart, respectively. The percentage inhibition (%) was calculated by the equation as follows:

%Inhibition={1–(Asam,10–Asam,0)÷(Acont,10–Acont,0)}×100.

Molecular docking study

Preparation of ligands and receptors

Three-dimensional (3D) structures of bisresorcinol (CID 8917124) and reference compounds, including substrates and inhibitors, were downloaded from PubChem database for molecular docking study. A ligand structure was prepared in a Protein Data Bank (PDB) format file using Online SMILES Translator and Structure File Generator (https://cactus.nci.nih.gov/translate/). The crystal structures of enzymes, such as collagenase (PDB ID 2Y6I), elastase (PDB ID 1BRU), and tyrosinase (PDB ID 2Y9X) were obtained from RCSB protein database (http://www.rcsb.org). The water and the attached molecules were dissected from all selected protein structures. Meanwhile, polar hydrogen atoms were added in the crystallized protein structures by AutoDockTools 1.5.6. Files of target proteins and other used compounds were saved in PDBQT format before performing the molecular docking.

Selection of active site residues and molecular docking

Grid and docking protocols of the active site predictions were prepared using AutoDockTools 1.5.6. Grid sites were set spacing of 0.375 Å. The x-y-z dimensions were set to be 126-135-160 Å3 for collagenase, 130-120-126 Å3 for elastase, and 80-80-80 Å3 for tyrosinase. Grid box centers (with offset values in AutoDockTools) were 31.869 (5.000), −19.41 (−25.000), and 17.815 for collagenase, 30.048 (−0.972), 51.253 (−2.722), 17.6 for elastase, and −8.407 (−1.000), −23.795 (−0.250), −36.019 (−3.500) for tyrosinase, respectively. The protein structure was used as a rigid entity while the ligand compound was set as a flexible molecule. Docking study was performed using the Lamarckian genetic algorithm (GA) implemented by AutoDock4 version 4.2. The number of GA runs was 50 with a popular size of 200. The binding energy (ΔGbind) was analyzed by ADT. Interaction(s) was visualized using BIOVIA Discovery Studio (BIOVIA, 2020). Molecular interaction between the compound and a protein receptor was analyzed and visualized using BIOVIA Discovery Studio software (BIOVIA, 2020). The structure of protein binding site compound was visualized using Visual Molecular Dynamics (VMD) package (Humphrey, Dalke & Schulten, 1996).

Statistical analysis

For in vitro study, each experiment was done in triplicate. Data were expressed as means ± standard error of the mean (SEM). The Student’s t-test was used to evaluate differences between results, and p-values of <0.01 were indicated to be significantly different. The performance regarding statistics was run on Libre Office Calc 5 in Ubuntu 16.04.6 LTS software.

Results

A major constituent was successfully purified from H. terminalis’s trunk by using solvent extraction and reverse phase HPLC techniques, respectively. The acquired 1H-NMR and 13C-NMR spectra were compared to those of reference substances through the SciFinder (https://scifinder.cas.org/) and found to correspond with (8′Z)-1,3-dihydroxy-5-[16′-(3″,5″-dihyroxyphenyl)-8′-hexadecen-1′-yl] benzene (Chaturvedula et al., 2002). It was classified as one of bisresorcinols. This recent bisresorcinol had the molecular formula of C28H40O4 with the molecular weight of 463.2819 Dalton. The extraction yield was calculated to be 0.086%, based on the trunk’ dried weight. Its purity was more than 98%, concerning the 1H and 13C-NMR spectra acquired (see Figs. S1 and S2).

In vitro bioactivity analysis

Antagonistic effects of the bisresorcinol on aging enzymes, such as collagenase, elastase, and tyrosinase were determined in vitro by using a distinct spectroscopic method. Results were summarized in Table 1 and Fig. 2. For collagenase inhibition, test samples in a range of 50–550 µmol L−1 were prepared using MOCAc-PRO-Leu-Gly-Leu-A2pr (Dnp)-Ala-Arg-NH2 as the enzyme substrate and caffeic acid as an enzyme inhibitor. It was indicated that the bisresorcinol at a concentration of 156.7 µmol L−1 decreased the enzyme activity by 50% (IC50). Instead, the IC50 of caffeic acid was 308.9 ± 13.7 µmol L−1. Thus, anti-collagenase activity of the bisresorcinol was significantly stronger than caffeic acid.

Figure 2 Response of inhibition percentage.

Response of % inhibition, evaluated by linear regression analysis and correlation coefficients expressed as R2 shown in the panels; (A) collagenase inhibition in compared to caffeic acid; (B) elastase inhibitory activity using ursolic acid as a positive control; and (C) tyrosinase inhibitory assay in comparison with β-arbutin.

Table 1 The IC50 values of the bisresorcinal and positive standards for enzymatic inhibitory assays regarding collagenase, elastase, and tyrosinase.

Assay	IC50 (µmole L−1 ± SEM)	
Collagenase inhibition	
Bisresorcinol	156.7 ± 2.3*	
Caffeic acid	308.9 ± 13.7*	
Elastase inhibition	
Bisresorcinal	33.2 ± 1.1	
Ursolic acid	34.3 ± 0.6	
Tyrosinase inhibition	
Bisresorcinol	22.6 ± 1.3**	
β-Arbutin	78.5 ± 3.1**	
Notes:

* p < 0.01.

** p < 0.001.

Inhibition of elastase enzyme was determined at a concentration range between 10 and 100 μmol L−1. The residual enzyme activity was spectroscopically measured at 410 nm in response to the amount of p-nitroaniline released from the substrate SANA after cleavage. The bisresorcinol strongly inhibited elastase activity in a concentration-dependent manner with the maximum inhibition of 100% at 50 µmol L−1. However, the inhibition decreased rapidly at concentrations less than 30 μmol L−1. Notably, the dose-response curve of ursolic acid was less steep than that of the bisresorcinol by reaching 80% inhibition at 100 µmol L−1. The IC50 values of the bisresorcinol and ursolic acid were respectively 33.2 and 34.3 µmol L−1, which were not significantly different.

In analysis of tyrosinase inhibition, samples were diluted to a concentration range of 10–100 µmol L−1. L-tyrosine was used as a substrate of mushroom tyrosinase and β-arbutin was a known inhibitor. The bisresorcinol showed the IC50 of 22.6 µmol L−1. The IC50 of β-arbutin was 78.5 µmol L−1. Our test compound showed 100% inhibition at 50 µmol L−1. In contrast, the enzyme inhibition of β-arbutin higher than 50% was not determined in the tested range.

Molecular interaction analysis

Molecular docking was applied to predict binding sites of test compounds to protein receptors, such as collagenase, elastase and tyrosinase, in comparison with known inhibitors of a corresponding enzyme (Supplemental Information). Consisting with previous publications, the relative binding affinity was evaluated meanwhile binding interactions were illustrated through the best predicted conformation (Teajaroen et al., 2020; Jewboonchu et al., 2020; Tanawattanasuntorn et al., 2020; Saeloh et al., 2017).

For collagenase enzyme, results of ligand-protein interactions were demonstrated in Fig. 3 and Table S1. The binding energy of −5.89 kcal mol−1 to clostridial collagenase (PDB ID 2Y6I) was presented by the bisresorcinol. A range of the binding energy between −3.68 and −7.90 kcal mol−1 was determined for other known collagenase inhibitors, including caffeic acid.

Figure 3 Collagense bound ligands.

(A) Molecular docking model of caffeic acid (red) and the bisresorcinol (blue) on the collagenase active site. Ligand interaction diagrams of caffeic acid (B–C) and the bisresorcinol (D–E), involving the π-π stacking, hydrogen bond, and Van der Waals as respectively depicted in pink, green, and grey.

Both the bisresorcinol and caffeic acid shared the collagenase binding sites (Fig. 3A). Four amino acids responsible for caffeic acid attachment included His524, Trp496, His527, and Trp539 (Fig. 3B). His524 and Trp496 interacted caffeic acid by hydrogen bonds through carboxylic and phenolic hydroxyl group(s). His527 and Trp539 bonded to the phenolic ring by π-π stacking. Other amino acids within the enzyme pocket contributed their binding through Van der Waals force (Fig. 3C). Interactions between the collagenase and the bisresorcinol were shown in the Figs. 3D and 3E, involving Trp496 and Trp539 amino acids that donated π-electrons to bond to the phenolic ring of the bisresorcinol. The presence of hydrogen bonds between the hydroxyl groups of the bisresorcinol and amino acids Asp601 and Ser602 was observed. Moreover, the Zn atom in the enzyme active site might be coordinated to both the bisresorcinol and caffeic acid.

The crystal structure of pig elastase (PDB ID 1BRU) was adopted to determine binding behaviors to elastase enzyme of the bisresorcinol and other known inhibitors, including ursolic acid. Results were shown in Fig. 4 and Table S2. The binding energy of all test compounds ranged between −5.62 and −8.94 kcal mol−1 and that of −5.69 kcal mol−1 was calculated for the bisresorcinol.

Figure 4 Elastase bound ligands.

(A) Binding characteristics of the bisresorcinol (blue) and ursolic acid (red) to the active site of pig elastase; (B–C), Ligand-protein interaction diagrams of ursolic acid (B–C) and the bisresorcinol (D–E); Interactions involved π-π stacking, hydrogen bonds, and Van der Waals force, depicted in pink, green, and grey, respectively.

There was similarity in binding to the elastase of the bisresorcinol and ursolic acid (Fig. 4A). Regarding ursolic acid, hydrogen bonding was proposed for a cyclic aliphatic hydroxyl group and Ser96, as well as a carboxylic group and Asn192 (Figs. 4B and 4C). Interestingly, structural folding of the bisresorcinol was presumed. This might facilitate interactions between phenolic hydroxyl groups of the compound and amino acids like Asn147, Ser190, Phe215, and Ser217 through hydrogen bonding (Figs. 4D and 4E).

To investigate binding affinity to tyrosinase of compounds, the crystal structure of mushroom tyrosinase (PDB ID 2Y9X) was applied. Results were displayed in Fig. 5 and Table S3, indicating that the binding energy of the bisresorcinol was −6.57 kcal mol−1, being in a range of −4.63 to −8.12 kcal mol−1 for other known inhibitors. In contrast, all known substrates were tightly bound to the tyrosinase according to an enormous decrease of the binding energy to a range from −15.46 and −23.94 kcal mol−1.

Figure 5 Tyrosinase bound ligands.

(A) Molecular docking model of β-arbutin (red) and the bisresorcinol (blue) on the tyrosinase active site. Ligand-protein interaction diagrams of β-arbutin (B–C) and the bisresorcinol (D–E) showed the engagement of π-cation, π-π stacking, hydrogen bonding, and van der waals force, depicted in orange, pink, green, and grey, respectively.

The coordinate covalent bonds between metal ions and β-arbutin or the bisresorcinol at the tyrosinase active site likely occurred. However, copper ions (Cu2+) were replaced by zinc ions (Zn2+) in the AutodockTool, because the force field for Cu2+ is unavailable and Cu2+ and Zn2+ ions are quite identical in charges and sizes (Santos-Martins et al., 2014). For β-arbutin, one zinc ion might covalently bond to the phenolic hydroxyl group, and another zinc ion interacted with the phenol ring through π-cation bond with His259 and His263 residues (Figs. 5B and 5C). In addition, the amino acids, such as Asn260, Ser282, and Val283, are linked to its hydroxyl groups by hydrogen bonding. In view of the bisresorcinol, the phenolic hydroxyl group formed hydrogen bonds with Met280, Ala 246, and Glu239. The π-π stacked interaction between the phenol ring and His263, and the π-cation attraction between another phenol ring and Arg321 were distinctly observed.

In summary, molecular interactions and binding affinity between the bisresorcinol and collagenase, elastase, or tyrosinase were tabulated in Table 2.

Table 2 Binding energy and amino acid residues that participated in the binding.

Ligand	Binding energy (kcal/mol)	Binding residue	
Collagenase inhibitor			
Caffeic acid	−6.86	Glu524, Tyr496,
His527, Trp539,
Glu498, Glu555, His523, Ile497, Leu495, Pro499	
Bisresorcinol	−5.89	Asp601, Glu498, Ser602,
Tyr496, Trp539,
Asp603, Gln530, Glu524, Glu555, Gly494, Ile497, Pro499, Thr551, Trp604, Tyr607, Ser597, Zn1791	
Elastase inhibitor			
Ursolic acid	−8.94	Asn192, Ser96,
Asn97, Gly216, Phe215, Ser190, Ser195, Ser214, Ser217, Thr146, Trp94, Val213	
Bisresorcinol	−5.69	Asn147, Phe215, Ser190, Ser217,
Asn192, Arg143, Gly216, Gly219, His57, Ser189, Ser195, Ser214, Ser226
Thr146, Val213, Val227	
Tyrosinase inhibitor			
β-arbutin	−5.52	His259, His263, Zn401
Asn260, Ser282, Val283,
His61, His85, His94, His296, Phe90, Phe264, Phe292, Pro284	
Bisresorcinol	−6.57	Arg321,
Ala246, Glu239, Met280,
His263,
Asn243, Asn260, Glu322, Gly86, Gly281, His61, His85, His94, His251, His259, His296, Met319, Phe292, Ser282, Thr87, Val248, Zn400	
Note:

π-cation, π-π stacking, and hydrogen bonds were depicted in italic, underline, and bold, respectively. The normal letter represented amino acids containing and Van der Waals interactions with the compound.

Discussion

Curcuminoid, a phenolic compound from Curcuma longa L., has been included in various cosmetic products as an antioxidant for anti-aging property (Gopinath & Karthikeyan, 2018). Since phenolic hydroxy determinants were existent, the bisresorcinol from H. terminalis’s trunk was anticipated to harbor anti-aging activity accordingly (Fig. 1). In the recent in vitro inhibitory assays, the bisresorcinol was found to dominantly inhibit elastase and tyrosinase activities rather than collagenase counterpart (Fig. 2). Accordingly, in silico docking experiments were performed to ascertain binding affinity of the bisresorcinol relevant to these aging enzymes. Interactions regarding π-electrons and hydrogen bonds were assumed to be key determinants for its bindings. Moreover, hydrophobic interactions that occurred between phenolic hydroxyl groups and amino acid residues at/nearby the enzyme active sites might be contributors for loss of enzyme functions, consisting with the previous literatures (Medvidović-Kosanović et al., 2010; Pientaweeratch, Panapisal & Tansirikongkol, 2016). Herein, binding characteristics of the bisresorcinol to each enzyme were distinctly explained based on structural comparison corresponding to a specific inhibitor. Inhibition of clostridial collagenase by caffeic acid, EGCG, quercetin, and catechin has been documented (Szewczyk et al., 2020; Pluemsamran, Onkoksoong & Panich, 2012; Hong et al., 2014). Similar to other known inhibitors, the hydroxy phenol groups of the bisresorcinol were suggested to interact with key amino acids of the collagenase binding site through π-π interaction and hydrogen bonding. Interestingly, the long flexible structure of bisresorcinol had a capability to fold or elongate. Thus, when elongated, the unbound phenol ring on the other side could bind to amino acid residues at the active site, resulting in increased binding strength. By this way, inhibitory potential of the bisresorcinol on collagenase activity was proposed (Fig. 6).

Figure 6 Binding modes of collagenase inhibitors.

Binding modes of inhibitors namely caffeic acid, catechin, EGCG, and quercetin, and the bisresorcinol on collagenase enzyme; Symbols: red dash lines, hydrogen bonds; green dash lines, π-interactions.

In agreement with a previous study (Huang et al., 2013), binding to elastase enzyme of both the bisresorcinol and known inhibitors, such as procyanidin, quercetin and ursolic acid, were identical, involving hydrogen bond formation. Moreover, interactions between the bisresorcinol and amino acids apart from the enzyme pocket were speculated to mediate the unfolded phenol ring of the substance (Fig. 7).

Figure 7 Binding modes of elastase inhibitors.

Binding modes of actions of the bisresorcinol and known inhibitors, such as quercetin, procyanidin, and ursolic acid on elastase enzyme; Symbol: red dash lines, hydrogen bonds.

In view of tyrosinase inhibitory activity, inhibitors like kojic acid, rutin, and L-mimosine, have been apparent to competitively bind to tyrosinase enzyme with L-tyrosine, leading to inhibition of melanin synthesis (Channar et al., 2018; Nguyen & Tawata, 2015; Si et al., 2012). In Fig. 8, the π-π interactions between any test compound and histidine residues of the copper active site of tyrosinase were suggested. In agreement with a previous research, this particularly provided an explanation for other antagonistic effects on tyrosinase activity of such compounds (Lai et al., 2017). In fact, the phenolic skeleton in structures has been implicated in designing derivatives of indanone (Jung et al., 2019) and thiazolyl resorcinol (Mann et al., 2018), supporting the possibility of using the bisresorcinol as an antagonist of tyrosinase enzyme.

Figure 8 Binding modes of tyrosinase inhibitors.

Binding modes of actions of the bisresorcinol and known inhibitors, e.g., kojic acid, L-mimosine, and rutin, on mushroom tyrosinase; Symbols: red dash lines, hydrogen bonds; green dash lines, π-interactions.

Conclusion

The bisresorcinol might serve as a competitive inhibitor for collagenase, elastase, and tyrosinase with comparable modes of binding compared to known inhibitors. However, the long and flexible structure of the bisresorcinol was interesting in that additional interactions towards an unbound phenol ring to neighboring amino acids of the enzymes might be apparent. This finding was firstly reported and might give an idea for development of new cosmetic products containing the bisresorcinol for anti-aging and whitening effects. Nevertheless, there were further needs to examine its potency in vivo and by clinical trials.

Supplemental Information

Supplemental Information 1 1H NMR spectrum of bisresorcinol.

Click here for additional data file.

Supplemental Information 2 13C NMR spectrum of bisresorcinol.

Click here for additional data file.

Supplemental Information 3 Raw data for IC50 values of the bisresorcinal and positive standards for enzymatic inhibitory assays regarding collagenase, elastase, and tyrosinase.

Click here for additional data file.

Supplemental Information 4 Linear fitting for bioactivity of bisresorcinol.

Linear fitting came from the data from Table 1 in three anti-aging related enzymes.

Click here for additional data file.

Supplemental Information 5 Nonlinear fitting for bioactivity of bisresorcinol.

Nonlinear fitting came from the data from Table 1 in three anti-aging related enzymes.

Click here for additional data file.

The authors sincerely thank to other research facility providers, such as the Graduate School at Prince of Songkla University; Drug Delivery System Excellence Center, Faculty of Pharmaceutical Sciences, Prince of Songkla University; the Graduate School of Biomedical and Health Sciences (Pharmaceutical Sciences), Hiroshima University, Hiroshima, Japan; and Department of Biomedical Sciences and Biomedical Engineering, Faculty of Medicine, Prince of Songkla University, Thailand.

Additional Information and Declarations

Competing Interests

Author Contributions

Data Availability

The authors declare that they have no competing interests.

Charinrat Saechan conceived and designed the experiments, performed the experiments, analyzed the data, prepared figures and/or tables, authored or reviewed drafts of the paper, and approved the final draft.

Uyen Hoang Nguyen performed the experiments, analyzed the data, prepared figures and/or tables, and approved the final draft.

Zhichao Wang performed the experiments, analyzed the data, prepared figures and/or tables, and approved the final draft.

Sachiko Sugimoto conceived and designed the experiments, authored or reviewed drafts of the paper, and approved the final draft.

Yoshi Yamano conceived and designed the experiments, authored or reviewed drafts of the paper, and approved the final draft.

Katsuyoshi Matsunami conceived and designed the experiments, authored or reviewed drafts of the paper, and approved the final draft.

Hideaki Otsuka performed the experiments, authored or reviewed drafts of the paper, and approved the final draft.

Giang Minh Phan performed the experiments, authored or reviewed drafts of the paper, and approved the final draft.

Viet Hung Pham performed the experiments, authored or reviewed drafts of the paper, and approved the final draft.

Varomyalin Tipmanee conceived and designed the experiments, performed the experiments, analyzed the data, prepared figures and/or tables, authored or reviewed drafts of the paper, and approved the final draft.

Jasadee Kaewsrichan conceived and designed the experiments, analyzed the data, prepared figures and/or tables, authored or reviewed drafts of the paper, and approved the final draft.

The following information was supplied regarding data availability:

Raw data for IC50 values of the bisresorcinal and positive standards for enzymatic inhibitory assays regarding collagenase, elastase, and tyrosinase are available in the Supplemental Files.

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
