# Peer review of "Potency of bisresorcinol from Heliciopsis terminalis on skin aging: in vitro bioactivities and molecular interactions"

_PeerJ, doi:10.7717/peerj.11618_

## Round 0.1 · original submission · Minor Revisions

In this manuscript, the authors suggest bisresorcinol as a potential anti-aging substance by demonstrating the inhibitory activities against three enzymes including collagenase, elastase, and tyrosinase. The result is meaningful and interesting. However, the current paper is required to be revised actively according to the reviewers' comments before it could be considered for publication in PeerJ. In addition to the reviewers' comments, please have the data fit in the non-linear regression models shown in Figure 2. Also, please improve the resolution of the graph in Figure 2, if possible.

Reviewer 1 ·

Basic reporting

All terms and materials suitable for the subject have been provided.

Experimental design

A suitable experimental design was used to approach the experimental purpose.

Validity of the findings

It provides basic data that can be used as a cosmetic material for bisresorcinol.

Additional comments

This study confirmed the biological effects of bisresorcinol on tyrosinase, collagenase, and elastase in vitro. In addition, the interaction of bisresorcinol at the molecular level was confirmed using a docking technique.

1. The inhibition mechanism for various enzymes in vitro will not be linear regulation as shown in Figure 2. Since enzymes have a tertiary or quaternary structure, the mode of action with a substrate or an inhibitor proceeds by hyperbolic or sigmoidal reaction. How can linear regulation come about?
2. The raw data provided only conducted enzyme inhibition experiments on three concentrations, and how did Figure 2 show more than three concentrations?
3. What is the yield of bisresorcinol extracted from Heliciopsis terminalis, and what is the purity of bisresorcinol used in the experiment? Please refer to this information in the paper or provide it as a supplementary material.

Reviewer 2 ·

Basic reporting

BASIC REPORTING
The title, abstract, introduction, methods, results and discussion are appropriate for the content of the text. Furthermore, the article is well constructed, the experiments are well conducted, and analysis is well performed. The figures are relevant, high quality, well labelled and described.

Experimental design

EXPERIMENTAL DESIGN
The experimental design is original and the research is within the scope of the journal. Research question is well defined, relevant and meaningful. The methods are highly technical, ethical and logistical.

Validity of the findings

VALIDITY OF THE FINDINGS
All underlying data have been provided in detail. The findings are meaningful. The conclusions are well stated and relevant to the research questions.

Additional comments

This paper demonstrated the function of bisresorcinol purified from H. terminalis in skin aging. The authors investigate its inhibitory activity in vitro on collagenase, elastase, and tyrosinase using a series of spectroscopic techniques. The molecular interaction analysis was further evaluated using molecular dynamic simulation studies to decipher the binding sites of test compounds to collagenase, elastase and tyrosinase. In short, the authors demonstrate that the bisresorcinol purified from H. terminalis might be useful for inclusion in cosmetic products as an aging-enzyme antagonist.

Editorial Criteria
BASIC REPORTING
The title, abstract, introduction, methods, results and discussion are appropriate for the content of the text. Furthermore, the article is well constructed, the experiments are well conducted, and analysis is well performed. The figures are relevant, high quality, well labelled and described.
EXPERIMENTAL DESIGN
The experimental design is original and the research is within the scope of the journal. Research question is well defined, relevant and meaningful. The methods are highly technical, ethical and logistical.
VALIDITY OF THE FINDINGS
All underlying data have been provided in detail. The findings are meaningful. The conclusions are well stated and relevant to the research questions.

Overall, I think this paper is novel and will be of interest to others in the community of skin anti-aging strategies research. The statistical part is valid and makes sense. The authors make it comprehensive by investigating the molecular mechanisms behind its anti skin aging function in a qualitative manner. It can be assumed that the in vitro assays performed with the collagenase, elastase, tyrosinase and respected synthetic peptides are reliable tool to get first indications of the capability of bisresorcinol from Heliciopsis terminalis on skin aging. In general, the work is convincing except some major and minor comments below:


Major Comments:

I’m wondering if the cells used in culture were routinely subjected to mycoplasma testing or not? If so, please add sententense like “only cells that were negative for mycoplasma were used for experiments” in the Methods.

It’s better to add “all the chemicals used in this study were of analytical grade” or similar idea in the Chemicals and Instruments part.




Minor Comments:
The authors didn’t mention what tools were used to perform molecular interaction visualization for the selected receptors and ligand.

Annotated reviews are not available for download in order to protect the identity of reviewers who chose to remain anonymous.

---

## Round 0.2 · accepted · Accept

The manuscript has been revised according to the reviewers' comments. I appreciate all your effort to address the answers. We are pleased to accept your paper in its current form.

Reviewer 2 ·

Basic reporting

The title, abstract, introduction, methods, results and discussion are appropriate for the content of the text. Furthermore, the article is well constructed, the experiments are well conducted, and analysis is well performed. The figures are relevant, high quality, well labelled and described.

Experimental design

The experimental design is original and the research is within the scope of the journal. Research question is well defined, relevant and meaningful. The methods are highly technical, ethical and logistical.

Validity of the findings

All underlying data have been provided in detail. The findings are meaningful. The conclusions are well stated and relevant to the research questions.

Additional comments

This paper demonstrated the function of bisresorcinol purified from H. terminalis in skin aging. The authors investigate its inhibitory activity in vitro on collagenase, elastase, and tyrosinase using a series of spectroscopic techniques. The molecular interaction analysis was further evaluated using molecular dynamic simulation studies to decipher the binding sites of test compounds to collagenase, elastase and tyrosinase. In short, the authors demonstrate that the bisresorcinol purified from H. terminalis might be useful for inclusion in cosmetic products as an aging-enzyme antagonist.

Overall, I think the corrections look good to me. All my comments have been answered in a proper manner. Please find the original comments and feedbacks below:

Editorial Criteria
BASIC REPORTING
The title, abstract, introduction, methods, results and discussion are appropriate for the content of the text. Furthermore, the article is well constructed, the experiments are well conducted, and analysis is well performed. The figures are relevant, high quality, well labelled and described.
EXPERIMENTAL DESIGN
The experimental design is original and the research is within the scope of the journal. Research question is well defined, relevant and meaningful. The methods are highly technical, ethical and logistical.
VALIDITY OF THE FINDINGS
All underlying data have been provided in detail. The findings are meaningful. The conclusions are well stated and relevant to the research questions.

Overall, I think this paper is novel and will be of interest to others in the community of skin anti-aging strategies research. The statistical part is valid and makes sense. The authors make it comprehensive by investigating the molecular mechanisms behind its anti skin aging function in a qualitative manner. It can be assumed that the in vitro assays performed with the collagenase, elastase, tyrosinase and respected synthetic peptides are reliable tools to get first indications of the capability of bisresorcinol from Heliciopsis terminalis on skin aging. In general, the work is convincing except some major and minor comments below:


Major Comments:

I’m wondering if the cells used in culture were routinely subjected to mycoplasma testing or not? If so, please add sententense like “only cells that were negative for mycoplasma were used for experiments” in the Methods.

Response:
Thanks for your suggestion. The word we used “in vitro'' could be misunderstood, because cell culture was not performed in this study. The mentioned “in-vitro” experiments meant enzyme-inhibitory assay only. To avoid such misunderstanding, we have added the sentence “In vitro experiments that consisted of enzyme-inhibitory assays on collagenase, elastase and tyrosinase for the bisresorcinol and known inhibitors were carried out as follows.” Under the subtopic “Biological Activity Assays'' of the topic “Methods”

Feedback:
Thanks for the clarification. That makes sense to me.


It’s better to add “all the chemicals used in this study were of analytical grade” or similar idea in the Chemicals and Instruments part.

Response:
Thanks for the suggestion. We put the sentence “All the chemicals used in this study were of analytical grade.” in “Chemicals and Instruments” part.

Feedback:
It looks good to me. Thanks!



Minor Comments:
The authors didn’t mention what tools were used to perform molecular interaction visualization for the selected receptors and ligand.
Response:
Thanks for the suggestion. In this study, we have used two packages to perform molecular interaction visualization, which were BIOVIA Discovery Studio software and Visual Molecular Dynamics (VMD). We have added the sentence at the end of the method section, with a track change, as followed “Molecular interaction between the compound and a protein receptor was analyzed and visualized using BIOVIA Discovery Studio software (BIOVIA, 2020). The structure of the protein binding site compound was visualized using Visual Molecular Dynamics (VMD) package (Humphrey, Dalke & Schulten, 1996)” to respond to your comment. The additional reference for VMD was also added in the reference section.
Feedback:
That is very clear. Thanks for the clarification. And thanks for adding the citations.

Annotated reviews are not available for download in order to protect the identity of reviewers who chose to remain anonymous.